# COVID-19 Scientific Facts vs. Conspiracy Theories: Is Science Failing to Pass Its Message?

**DOI:** 10.3390/ijerph18126343

**Published:** 2021-06-11

**Authors:** Marios Constantinou, Antonios Kagialis, Maria Karekla

**Affiliations:** 1Department of Social Sciences, School of Humanities and Social Sciences, University of Nicosia, Nicosia 2417, Cyprus; kaganthony@gmail.com; 2Department of Psychology, University of Cyprus, P.O. Box 20537, Nicosia 1678, Cyprus; mkarekla@ucy.ac.cy

**Keywords:** COVID-19, conspiracy theories, psychological state, public health, social distancing, quarantine measures

## Abstract

Science may be failing to convince a significant number of people about COVID-19 scientific facts and needed public health measures. Individual and social factors are behind believing conspiracy theories. Adults (N = 1001) were asked to rate their beliefs in various conspiracy theories circulating in social media, rate their psychological distress relating to COVID-19, rate their trust in science to solve COVID-19 problems, and rate their willingness to adhere to measures regarding social distancing and quarantine. The findings showed conspiracy theories are widely believed and related to lower age, lower education, living in less densely populated areas, and lower income. Stronger conspiracy theory beliefs predicted science mistrust and unwillingness to adhere to public health measures. Psychological state was a strong predictor of conspiracy beliefs. Recommendations, stemming from the findings, for reducing such beliefs and better serving public health are discussed.

## 1. Introduction

In the first 12 days of April 2020, within the first 100 continuous posts of each day (total of 1200 posts in 12 days) on the Facebook news feed of the first author there were 24.62 (SD = 2.12) posts per day regarding COVID-19 related conspiracy theories or myths (e.g., COVID-19 was created for human population control). An average of 2.56 (SD = 1.01) of those conspiracy theory posts called for some form of action or uprising against the government measures taken for COVID-19. At the same time, many older conspiracy theories and myths were resurfacing on social media or even linked somehow to the current COVID-19 situation (e.g., 5G telephony, vaccination link to autism, etc.). Conspiracy theory beliefs are not a new pandemic phenomenon [1,2], yet they seem to have taken central stage during this pandemic. Early in the COVID-19 pandemic emergence, the WHO Director General postulated that we are “not just fighting an epidemic; we’re fighting an infodemic. Fake news spreads faster and more easily than this virus, and is just as dangerous” [3]. Subsequently, others have studied this phenomenon and raise similar concerns [4].

When ancient Greeks could not explain naturally occurring destructive events, such as devastating floods, earthquakes, and mass deaths by famine and illnesses, they would explain the inexplicable by creating myths and deities [5,6]. Similar divine and mythical explanations were reported by Egyptians, Mayas, Incas, and other ancient civilizations. A common function of myths was to alleviate the anxiety created by the unknown and pass responsibility on to deities in an attempt to feel that not all in life is random [7]. It appears that even today, despite the advent of science, when events are inexplicable or anxiety provoking, humans rush to rationalize and turn to myths and conspiracy theories to help alleviate fears and anxieties [8].

In the same way the ancients used myths to explain the world, we might suggest that in contemporary society conspiracy theories play the same kind of role. Conspiracy theories are commonly defined as beliefs that attempt to explicate uncommon, distressing events, including disease epidemics [9]. Conspiracy theories attribute the cause of unexplained events to human malevolence [10,11]. “Conspiracy theories flourish in times of crisis when people feel threatened, uncertain, and insecure” [2]. Increased anxiety and other mental health problems, such as depression, have been indeed linked to higher levels of conspiracy theory beliefs [12]. The need to rationalize inexplicable events and diffuse anxiety levels, lead individuals to believing in imagined (vs. factual) theories [9]. Therefore, not unlike the ancient Greeks, individuals today form myths and conspiracy theories that entail some truth (e.g., coronaviruses are just another family of viruses with flu like symptoms) and some fabricated components (e.g., this coronavirus is just a plot by governments to control us [13].

The COVID-19 pandemic is unlike most previous global pandemics (e.g., the Spanish Flu 1918–1920) simply because of the digitalization of communications. Today, humans are socially connected and continuously updated with real or fake (and often a combination of both) news. This virtual social connection helps conspiracy theories spread quickly around the globe [14]. Unfortunately, a large number of individuals choose to believe myths and conspiracy theories over scientific evidence. The internet and social media are also widely used as means to receive information regarding anxiety and other mental or physical health problems [15] or simply as a venue for stress-relief and support [16]. Thus, especially those who may be more prone to stress, a bombardment of conspiracy theories from social media may feed their fears, provide plausible solutions, and render them more susceptible to believing such myths and conspiracies.

Additional parameters beyond stress management are associated with conspiracy theory belief. For example, people believe questionable health-related information when the presenter appears credible, impressionable, and/or effectively utilizes electronic communications [17]. Impressionable information (even if fake) is more easily recorded to memory than factual information [18]. Believing in conspiracy theories is a combination of individual (e.g., personality traits and education [19] and social factors (e.g., with whom you “hang out” [20]. Such factors also include lower self-esteem, political cynicism, social exposure to conspiracy theories, lower agreeableness, generalized distorted thinking and paranormal beliefs, chronic feelings of psychological and sociopolitical disempowerment, intense feelings of uncertainty, lower educational level, lower crystallized intelligence, feelings of belonging in an intergroup social conflict, mistrust of government, paranoia/suspiciousness, lower analytical thinking, schizotypy, and stronger religiosity [1,9,19,21,22,23]. Interestingly, for some, a conspiracy theory is more appealing and satisfying than factual information [24].

The existence and intensity of beliefs in conspiracy theories becomes a social and individual health and safety risk when such beliefs lead people to act against their own and/or others’ best interests. For example, measles is resurging due to the widespread conspiracy theory beliefs surrounding vaccinations [23]. Studies are reporting that refusing empirically supported medications and lifesaving vaccinations, rejecting the use of sunscreens, and turning to unsupported alternative medicine and rejecting the need for physical check-ups, are often linked to conspiracy theory beliefs [25]. Recently, [26] in a study reported alarming findings that showed that untrue rumors and conspiracy theories are leading people to vaccine hesitancy for COVID-19.

Past research has not studied in detail whether conspiracy theory beliefs place health measures in danger and whether believers of such theories are at a higher risk of defying such health measures. In addition, it is not clear from the literature if conspiracy beliefs are associated with greater stress during a time of crisis, i.e., the first wave of the COVID-19 pandemic. This study aimed to ascertain the prevalence of conspiracy beliefs in the first half of April 2020, during strictly enforced social isolation and quarantine regulations. Secondly, it explored the impact of such beliefs on public health and safety behaviors (i.e., adherence to governmental regulations and recommendations) by assessing whether conspiracy beliefs predict willingness to adhere to such measures. Thirdly, the study assessed whether the subjective reporting of psychological distress relating to COVID-19 predicts higher levels of conspiracy theory beliefs and lower adherence to social distancing and health recommendations. The study also evaluated the relationship between trust in scientists and believing conspiracy theories.

In line with Cherry [23], it was hypothesized that believing in conspiracy theories is related to lower willingness to adhere to enforced measures aimed at maintaining public health, such as social distancing and quarantines. It was also hypothesized, in line with [12], that higher subjective reports of psychological distress due to COVID-19 would be predictive of higher conspiracy theory beliefs. Finally, in line with past studies, which found that more impressionable fake explanations are more likely to be followed than factual information [24], trust in scientists was hypothesized to be lower in conspiracy theory believers.

## 2. Methods

### 2.1. Participants

The study was conducted in Cyprus and Greece with 42.90% of respondents residing in Greece. An opportunistic sample of 1001 individuals (18 years or older; 80.50% women) participated during the week that the questionnaire remained posted online. Participants’ average age was 35.59 years (*SD* = 10.07, range = 19–73 years). Ninety-three percent of the participants had at least a bachelor’s degree. Sixty percent lived in an urban setting (city with more than 100,000 inhabitants) and the median individual income was EUR 995 per month (due to a large number of outliers on both ends the mean was deemed nonrepresentative) with the lowest individual income being zero and the highest EUR 8720 per month.

### 2.2. Procedure

The Cyprus National Bioethics Committee approved the study. Invitation calls for the study were posted online via Facebook and Twitter and also emailed to personal contacts and university professors to share on their social media sites. The study did not receive any funding and does not have any conflict of interest to report. The material and data are available upon request.

The study was completed about a month after the first measures of self-isolation and quarantine were enforced. Participation was open for seven days in April 2020, during which social distancing and quarantine measures were stringent >90/100 stringency score [27], and enforced with monetary punishments for offenders in both countries and an increasing curve of COVID-19 incidences being recorded. Interested individuals who provided electronic informed consent then completed a 10-minute-long internet-based questionnaire (in Google Forms).

### 2.3. Measures

Participants first provided demographic information (living area, personal income, age, sex, and education) and then completed the following study measures.

Conspiracy Theory Beliefs. Participants were asked to respond on a Likert scale (1–10; with 1 = “certainly no” and 10 = “certainly yes”) how strongly they believe eight statements related to COVID-19 and one statement relating to general beliefs in conspiracy theories (Table 1). The statements were created by the authors, who agreed upon popular conspiracy theories circulating on social media at the time regarding COVID-19.

Likelihood of adhering to governmental regulations imposed as a result of the COVID-19 pandemic. The willingness to adhere to social distancing and quarantine governmental regulations was rated on a 10-point Likert scale (1 = least likely to 10 = most likely to follow recommendations) in two questions: “I will adhere to the mandated measures” and “I will adhere to the mandated measures for as long as it takes.” **Trust in science** was similarly rated on a 1–10 scale with 1 = lowest trust to 10 = complete trust. Participants were also asked to rate their trust in science/scientists in relation to COVID-19 with two statements, “I believe science is useful for solving the COVID-19 problem” and “only science can solve the COVID-19 problem.”

COVID-19 psychological state. Participants rated on a Likert scale (1–10; with 1 = “certainly no” and 10 = “certainly yes”) their present-moment subjective feelings of distress, hopelessness, sadness, and being on edge, all of which were assessed with single-item responses, in an attempt to keep the questionnaire short and maximize participant completion. Single-item screening tools for anxiety, melancholy, and stress have been used in the past and found to be sensitive for screening purposes [28,29,30].

The complete questionnaire (in Greek) is available from the authors upon request.

## 3. Results

In order to evaluate the prevalence of beliefs (first aim of the study) in each of the nine conspiracy theory statements, three percentile groupings (quartiles) were calculated as follows: (1) 1–25th percentile, (2) 26th to 75th percentile, and (3) 76th to 99th percentile, which represented none-to-weak belief, moderate belief, and strong belief, respectively (see Table 1). Subsequently, the percentage of the 1001 participants belonging in each of the three groups was calculated and reported in Table 1.

A factor analysis (principal component analysis, two-tailed) of the nine conspiracy theory statements produced only one factor (conspiracy beliefs, CB) with 51.33% of the variance explained. With a sample of 1001 participants, the critical value for loading on a factor is suggested to be 0.162 [31]; all nine statements loaded strongly on the one factor (see Table 1). The internal reliability (listwise deletion) of the nine conspiracy theory items was high, Cronbach’s alpha = 0.89. Thus, the total score (CB) of all nine statements was calculated and treated as one variable in regression analyses.

In order to evaluate the need for entering covariates in further statistical analyses, the correlations between the total CB score and sex (dichotomous; point biserial), age (continuous; Pearson r), income (continuous; Pearson r), living area (continuous measured with size town; Pearson r), and education (continuous) were calculated (Table 2).

Two statements assessed the willingness of individuals to adhere to scientifically recommended and government mandated measures (i.e., social distancing and quarantine). These statements were highly correlated (Pearson *r* = 0.85) and were treated, for further analyses, as one total variable: adherence behavior (AB). The correlations between AB and demographics are presented in Table 2.

A principal component factor analysis with the four items assessing subjective feelings of COVID-19 related to distress, hopelessness, sadness, and being on edge, proposed only one factor (psychological state; PS), 68.02% of the variance explained, with each item loading strongly (i.e., distress = 0.79, sadness = 0.88, hopelessness = 0.82, and being on edge = 0.80). Cronbach’s alpha was high and equal 0.84, suggesting internal consistency of these items. The scores on these statements were thus summed to form a COVID-19 psychological state scale with higher scores indicating greater psychological distress at present and relation to the COVID-19 pandemic. The correlations between PS and demographics are reported in Table 2.

Participants rated their trust in science/scientists in relation to COVID-19 with two statements. These two statements were moderately correlated (Pearson *r* = 0.51, *p* < 0.01), and a new variable, total trust in science (TS), was calculated. The correlations between TS and demographics are presented in Table 2.

### 3.1. Conspiracy Theory Belief and Adherence Behavior

The second aim of the study was to examine whether higher levels of conspiracy beliefs (CB) would predict lower adherence behavior (AB) to COVID-19 measures imposed by the governments. The linear regression assumptions were tested as follows: (a) the observations were independent (Durbin–Watson Statistic = 1.95); (b) the relationship between the two variables was significantly negative (Table 2) and linear; (c) the plotted residuals were approximately normally distributed; (d) the scatterplot of standardized residuals against standardized predicted values showed no discernible pattern and the assumption of homoscedasticity was met; and (e) no observations had a large influence (mean Cook’s distance = 0.001, SD = 0.006; mean centered leverage = 0.001, SD = 0.001).

The regression was significant: F_(1,999)_ = 16.77, *p* < 0.001, with R^2^ of 0.02 and standard error of the estimate being 2.93. The standardized beta for CB was −0.33, *t* = −11.05, *p* < 0.001. The regression equation was AB = 19.98 − 0.04 × CB. Thus, AB decreased 0.04 points for every point increase in CB.

In a hierarchical multiple regression, demographics (sex, age, education, living area, and income) were entered along with CB as independent variables. The highest VIF was equal to 1.24 and the highest tolerance was 0.98, thus meeting the collinearity assumption, see Table 2). When compared to the simple linear regression given above, the new R^2^ (0.12) improved significantly, *p* < 0.01, with the multiple regression being significant F_(5,995)_ = 27.78, *p* < 0.001, and its equation being AB = 18.23 − 0.04 × CB 0.03 × age. The other demographics did not have a significant (slope) input in the prediction equation. Running a multiple regression with only CB and age as independent variables did not significantly improve the model and R^2^ of the previous multiple regression.

### 3.2. Conspiracy Theory Beliefs and Psychological State

The third aim of the study was to evaluate whether the subjective report of psychological state (PS), due to COVID-19, was predictive of stronger conspiracy beliefs (CB). The linear regression assumptions were met as follows: (a) the observations were independent (Durbin–Watson statistic = 1.37); (b) the relationship between PS and CB was significant and linear at observation (*r* = 0.13, *p* < 0.001); (c) the plotted residuals were approximately normally distributed; (d) the plot of standardized residuals against standardized predicted values showed no discernible pattern and the assumption of homoscedasticity was met; and (e) no observations showed a large influence (mean Cook’s distance = 0.001, SD = 0.002; mean centered leverage = 0.001, SD = 0.001).

The regression was significant, F_(1,999)_ = 16.78, *p* < 0.001, with an R^2^ of 0.02, standard error of the estimate = 24.31. The standardized beta for PS was 0.13, *t* = 4.10, *p* < 0.001. The participants’ predicted CB was equal to 37.64 + 0.35 × PS. Thus, CB increased 0.35 points for each point increase in PS (distress).

In a hierarchical multiple regression, demographics (sex, age, education, living area, and income) were entered along with CB as independent variables. None of the VIF of variables was above 1.22 (well below 5) and the highest tolerance was 0.99, thus meeting the collinearity assumption (in addition, among the variables, none correlated highly, see Table 2). When compared to the simple linear regression, the new R^2^ (0.11) improved significantly, *p* < 0.001, with all of the demographic slopes being significant at *p* < 0.001 and the CB predicted by 101.87 + 0.30 × PS − 0.27 × (age) − 2.59 × (living area) − 5.22 × (education) − 0.03 × (income).

### 3.3. Conspiracy Theory Beliefs and Trust in Science

The final target of the study was to examine the hypothesis that stronger conspiracy beliefs (CB) would be predictive of lower trust in science (TS) to solve the COVID-19 problem. The assumptions for the linear regression mode were met as follows: (a) the observations were independent (Durbin–Watson statistic = 1.92); (b) there was a significant linear relationship between the two variables (*r* = 0.47); (c) the plotted residuals were approximately normally distributed; (d) the plot of standardized residuals against standardized predicted values showed no discernible pattern and the assumption of homoscedasticity was met; and (e) no observations showed a large influence (mean Cook’s distance = 0.001, SD = 0.004; mean centered leverage = 0.001, SD = 0.001).

The regression was significant, F_(1,999)_ = 277.83, *p* < 0.001, with an R^2^ of 0.22, standardized B = −0.47, *t* = −16.67, *p* < 0.001. The participants’ predicted TS was equal to 20.53 − 0.06 (CB), thus TS decreased by 0.06 points for every point increase in CB.

In a hierarchical multiple regression, demographics (sex, age, education, living area, and income) were entered along with CB as independent variables. The VIF was equal to 1.24 and tolerance was 0.97, thus meeting the collinearity assumption. When compared to the simple linear regression, the new R^2^ (0.22) did not improve significantly, *p* > 0.05, with most of the demographic slopes being insignificant at the *p* > 0.05, apart from the living area slope, *p* < 0.05, and the CB predicted by 21.60 − 0.06 × TS − 0.06 × (living area). Running a multiple regression with only CB and living area as the independent variables did not improve the R^2^ of the original simple linear regression.

## 4. Discussion

The first aim of this study was to examine the prevalence of common conspiracy theories circulating on social media during the social distancing and lockdowns for COVID-19 in April 2020. The reported percentages of beliefs in such theories (Table 1) are alarming, with about half of the sample strongly believing that “there is already a vaccine for COVID-19 and will be released when millions are infected”. The prevalence rates of the current study (about 20–50% strong beliefs, depending on the belief) are similarly high and alarming as the prevalence reported in several countries in a global survey [32], a study in the UK [33], and a study with German speakers [34].

Similar to Allington [35,36], who has been warning about the risks of conspiracy theories for the greater public, the current study demonstrated that as the strength of conspiracy theory beliefs increases, the willingness to adhere to public health recommendations and government enforced measures (in this instance for COVID-19) decreases significantly. At the same time, conspiracy theory believers were less trustful of science and scientists and therefore less likely to follow COVID-19 recommendations and measures. The seriousness of these findings can be further appreciated, when combined with past studies, which found rejection of scientific facts due to conspiracy theories “criminalizing” medicine and medications [37]; for example, avoiding vaccinations to the point that eradicated illnesses are returning, avoiding the use of condoms and placing individuals at higher risk for contracting HIV [38], and most recently avoiding vaccinations for COVID-19 [26]. During this pandemic, we witnessed citizen groups pushing for uprising and asking others to join them in breaching the “stay-at-home” and “social distancing” regulations enacted by governments and strongly recommended by scientists. Indeed, it has been found that conspiracy theory believers are more prone to aggression and stirring social unrest [39]. The call for uprising is often based on and/or strengthened by conspiracy theories and uprisers, who may endanger themselves and others in their community. Thus, it is evident that conspiracy theory belief is not merely a benign phenomenon [40]. Related to this phenomenon is psychological state, which was found to be a predictor of stronger conspiracy theory beliefs. Age, education, living area, and income significantly improve this prediction, with lower age, lower education, living in less densely populated areas, and lower income were also associated with stronger conspiracy theory beliefs.

This is a cross sectional self-report study and as such presents with several limitations. Participants constituted an opportunistic sample of social media users in Cyprus and Greece. Interestingly, the sample comprised mostly females with most being highly educated (tertiary education). As such, this may have introduced some bias with highly educated individuals being over-represented in this sample. However, as per the latest Eurostat “Educational Attainment Statistics,” Cyprus has the highest educational tertiary educational attainment in the European Union (about 60%) with women particularly presenting really high educational attainment rates [41]. Greece is also above the EU average for tertiary educational attainment [41]. Interestingly, despite the finding that educational attainment was inversely related to conspiracy theory beliefs, this sample of albeit highly educated individuals presented with high levels of conspiracy theory beliefs. This finding needs to be further explored in the future to examine whether and how science is failing to pass its message to the public.

Recommendations, stemming from the findings, can be made with the aim to reduce conspiracy theory beliefs and better serve public health. Easily accessible psychological support for people coming from all socioeconomic levels could have a large role in combating conspiracy theories or alleviating the need to utilize conspiracies to explain situations that may cause distress. At the same time, health and mental health professionals can affect conspiracy theory belief by offering reliable and valid psychoeducation surrounding COVID-19 related matters along with their other services. In particular, health professionals need to be constantly updated not only about valid scientific facts, but also concerning circulating internet and media-based fake news, myths, and conspiracy theories; by doing so, health professionals can join forces together in the fight against myths and conspiracy theories relating to COVID-19 and other pathogens. Grimes [42] recently proposed that health professionals and scientists should together aim to utilize education in the fight against conspiracy theory beliefs. For psychoeducation to be effective, it should also instigate analytical thinking, which appears to decrease conspiracy theory adherence [22].

The direct competition science has with conspiracy theories is intense. Conspiracy theories about COVID-19 are spreading faster than other ideas [4]. Is science failing to communicate effectively its messages about COVID-19? Are conspiracy theories winning over science? Unfortunately, our findings support that this may be the case. Conspiracies, in opposition to scientific jargon, appeal to emotions, tell interesting stories, and use simple, comprehensible, and impressionable language. Science can borrow from the impressionable methods used by conspiracy theorists and myth-spreaders. Science and its findings need to be delivered in more receptive and emotion-inducing ways. Science tends to present numbers, jargon, and difficult concepts to comprehend, even when informing the public in the news. For instance, even in prime-time news broadcasts scientists speak about COVID-19 and use jargon and acronyms such as RNA virus, ACE2 receptor binding, interaction with angiotensin, virus lipid membrane, glycoproteins, etc., without comprehensible explanations. Science could invest in science communication or the sociology of science so as to more effectively communicate the work of scientists to the public. For example, in line with the recent suggestions by the World Health Organization [3] and a recent study [26], conspiracy theories about COVID-19 and other public health concerns need to be combated immediately when they appear, and science could use social media to inform the public about correct facts. In addition, scientific findings need to be presented in simple, easily understood, comprehensible language. Science can also borrow techniques from marketing practices and from consumer behavior research [17,43] concerning capturing the attention of the public when it comes to public health issues and population safety. Scientists may consider themselves “flooding” social media and the internet, in general, with scientific facts and findings that help alleviate fears rather than elevate them. In fact, [40], proposed that societies, science, and social media need to be much more aggressive in blocking conspiracy theories. For these actions all sciences (e.g., medical, biological, social, educational, political), in coordination with local and international health organizations, need to work side by side. Such a proactive approach may help science win over conspiracy theory beliefs.

## Figures and Tables

**Table 1 ijerph-18-06343-t001:** Average responses, factor loadings, and percentage of weak, moderate, and strong belief endorsements for each conspiracy theory statement.

How Strongly Do You Believe Each of the Following Statements:	Avg (SD)	SEM	Factor CB Loading	% No-to-Weak Belief	% Moderate Belief	% Strong Belief
1. COVID-19 is not real	1.99 (1.75)	0.06	0.620	63.70	12.70	23.60
2. There is already a vaccine for COVID-19 and will be released when millions are infected	4.27 (2.83)	0.90	0.815	35.80	16.90	47.30
3. Deaths from COVID-19 in Italy, Spain, and USA are not as many as reported	3.48 (2.73)	0.09	0.611	36.20	40.50	23.30
4. Nobody really died from COVID-19	2.24 (2.53)	0.08	0.382	72.00	16.70	11.30
5. People dying from COVID-19 would have died very soon, anyway	3.05 (2.51)	0.08	0.539	45.80	36.00	18.20
6. I am generally a believer of conspiracy theories	3.55 (2.70)	0.09	0.794	33.70	43.60	22.70
7. With COVID-19 vaccinations we will be microchipped unwillingly	3.78 (2.97)	0.09	0.786	34.50	40.90	23.60
8. COVID-19 was created for population control	4.37 (3.01)	0.10	0.856	26.30	54.90	18.80
9. COVID-19 was created on purpose in a laboratory by scientists	5.20 (3.04)	0.10	0.781	26.6	29.90	43.50

Note: Avg = average on Likert scale 1–10 where 1 = «Certainly No» to 10 = «Certainly Yes»; SEM = standard error of the mean; Factor CB Loading = loading on factor “Conspiracy Belief; % No-to-Weak Belief = percentage of sample in 1st to 25th %ile; Moderate Belief = 26th to 75th %ile; and Strong Belief= 76th to 99th %ile.

**Table 2 ijerph-18-06343-t002:** Correlations between variables.

	CB	AB	PS	TS	Sex	Age	Education	Living Area
AB	−0.33 **							
PS	0.13 **	0.05						
TS	−0.47 **	0.33 **	0.01					
Sex	−0.10 **	−0.14 **	−0.13 **	0.06				
Age	−0.14 **	0.14 **	−0.03	0.10 **	0.08 *			
Education	−0.19 **	0.10 *	−0.06 *	0.06	−0.05	–0.04		
Living Area	−0.17 **	0.15 **	−0.04	0.02	0.03	0.07 *	0.04	
Income	−0.18 **	0.10 *	−0.01	0.09 *	0.14 **	0.39 **	0.17 **	0.05

Note: CB = conspiracy belief, AB = adherence behavior, PS = psychological state, and TS = trust in science; * *p* < 0.05 and ** *p* < 0.01.

## Data Availability

Data are available uppon request from the first author.

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
