# Peer review of "COVID-19 Scientific Facts vs. Conspiracy Theories: Is Science Failing to Pass Its Message?"

_ijerph, 2021, doi:10.3390/ijerph18126343_

Round 1

Reviewer 1 Report

This is a potentially interesting paper, but it needs work.

Another thing: on the first page the authors' write:

"When ancient Greeks could not explain naturally occurring destructive events, such as devastating floods, earthquakes, and mass deaths by famine and illnesses, they would explain the inexplicable by creating myths and deities. A common function of myths was to alleviate the anxiety created by the unknown and pass responsibility on to deities in an attempt to feel that not all in life is random."

This is commonly believed, but is it something that is actually argued by historians or the sociologists of history? It is necessary to cite someone here in those fields who makes this claim, rather than appeal to what might just be a myth about ancient mythology.

This is particularly important, because the authors then go on to link conspiracy theories with these "myths which alleviate anxiety." If the former claim about myths is erroneous, then linking conspiracy theories to myths is problematic. Compounding this issue is the fact that the authors never really define what they mean by "conspiracy theory." Instead, they assume that the reader will think that they are just examples of bad or irrational beliefs. However, academics who will be interested in this paper will doubtlessly be aware that there is some dicussion in the existing literature about how we should portray conspiracy theories. For example, Dentith, in their 2018 chapter in Joe Uscinski's book "Conspiracy theories and the people who believe them" argues that there are a variety of wats to define what counts as a "conspiracy theory" and the choice of definition affects the kind of analysis academics then run on belief in such theories. Thus, the authors need to define this term in order to situate them into the wider debate.

My other rationale for requesting a revision is that in order to review this paper properly, the questionaire and method used to disseminate the questionaire used to generate the survey results needs to be provided. As it stands, I cannot judge the quality of the results without seeing how the questions asked in said survey. This is especially the case because there is not enough said in the article about how research participants were picked out. According to the paper 93% of the participants had at least one unversity qualification; this seems awfully high and suggests the survey is somewhat skewed to one part of the population. The authors need to provide more information about the survey and the questionaire.

Finally, the editorialising about what to do about belief in conspiracy theories on page 7 does not connect well to the rest of the article. I understand where the authors are coming from, but unless the authors are mental health professionals, their recommendations for what mental health practioners should do seems overly simplistic.

Not just this, but in a subsequent paragraph the authors state:

"Science should instead be presenting a story of what its scientific facts mean for an individual, their relatives, their city, and their country. Science, in other words, should heavily invest in marketing and learning from consumer behavior research."

This is not an issue with science. This is an issue with science communication or the sociology of science. This is simply bad advice for scientists. Move this discussion on to something more about how we can more effectively communicate the work of scientists to the public, rather than chastise scientists for using appropriate terminology when engaging in scientific study.

In short: the term "conspiracy theory" needs defining in this paper. The authors also need to do a lot more to describe the survey method, and produce the questionaire (or, at least, representative sample questions from it). Finally, the authors need to examine some of the claims they make in the article and explain their import more fully.

Some other notes:

Page 2: In the paragraph starting "Additional parameters beyond stress management..." the authors mention factors which might influence or contribute to belief in conspiracy theories. Yet most, if not all, of these factors are also factors leading towards most political beliefs which aren't conspiratorial, so why pick out conspiracy theories as a class of belief here?

Author Response

On behalf of the author team, I would like to thank you for giving us the opportunity to revise and resubmit our work. We would also like to thank the reviewers for their comments, which we believe have greatly improved our manuscript.

Below you can find a detailed response to all reviewer’s comments.

Thank you.

Response to reviewers

Reviewer 1

  • On the first page the authors' write: "When ancient Greeks could not explain naturally occurring destructive events, such as devastating floods, earthquakes, and mass deaths by famine and illnesses, they would explain the inexplicable by creating myths and deities. A common function of myths was to alleviate the anxiety created by the unknown and pass responsibility on to deities in an attempt to feel that not all in life is random."

This is commonly believed, but is it something that is actually argued by historians or the sociologists of history? It is necessary to cite someone here in those fields who makes this claim, rather than appeal to what might just be a myth about ancient mythology. This is particularly important, because the authors then go on to link conspiracy theories with these "myths which alleviate anxiety." If the former claim about myths is erroneous, then linking conspiracy theories to myths is problematic.

We thank the reviewer for their comment. We have added references that support this claim that ancient myths tended to have the function of explaining what was otherwise impossible to understand or be easily communicated. Given this, we make the claim that myths were used as means to explain what was happening in individuals’ lives (e.g., natural phenomena) so as to alleviate anxiety and be able to function. We have included citations that have previously made similar claims.

  • Compounding this issue is the fact that the authors never really define what they mean by "conspiracy theory." Instead, they assume that the reader will think that they are just examples of bad or irrational beliefs. However, academics who will be interested in this paper will doubtlessly be aware that there is some discussion in the existing literature about how we should portray conspiracy theories. For example, Dentith, in their 2018 chapter in Joe Uscinski's book "Conspiracy theories and the people who believe them" argues that there are a variety of wats to define what counts as a "conspiracy theory" and the choice of definition affects the kind of analysis academics then run on belief in such theories. Thus, the authors need to define this term in order to situate them into the wider debate.

The reviewer is correct and this was an oversight. We have now defined the term conspiracy theory beliefs (see second paragraph of introduction) describing it in the way we conceptualized and measured it in this study while linking it with theories and researchers on the topic.

  • My other rationale for requesting a revision is that in order to review this paper properly, the questionnaire and method used to disseminate the questionnaire used to generate the survey results needs to be provided. As it stands, I cannot judge the quality of the results without seeing how the questions asked in said survey. This is especially the case because there is not enough said in the article about how research participants were picked out. According to the paper 93% of the participants had at least one university qualification; this seems awfully high and suggests the survey is somewhat skewed to one part of the population. The authors need to provide more information about the survey and the questionnaire.

The questionnaire was in Greek language and we guess it cannot be included as an appendix, however it is available per request. The methods section was revised to provide more information about the survey and measures. We clarified the question with its statements regarding conspiracy theory beliefs and provided the response options. All conspiracy theory belief statements (translated from the Greek language) are provided in Table 1.

Regarding the level of education of participants, Eurostat educational attainment statistics (April 2020) note that Cypriots that comprised the majority of the sample, have the highest tertiary education attainment in the EU with 60% of Cypriots having at least a Bachelors’ degree and women reaching considerably higher attainment rates than men. Greeks (who comprised the rest of the sample) also present via the same report above the European average attainment in tertiary education. The sample was an opportunistic one, comprised of individuals who use social media. This is presented in the methods section and discussed in the limitations of the study. We also changed the title so as to not overly emphasize this point.

  • Finally, the editorializing about what to do about belief in conspiracy theories on page 7 does not connect well to the rest of the article. I understand where the authors are coming from, but unless the authors are mental health professionals, their recommendations for what mental health practitioners should do seems overly simplistic.

The authors are mental health professionals (licensed clinical psychologists) and may have as such been biased in their attempt to provide recommendations based on their findings and expertise in order to help other health and mental health professionals and policy makers deal with this dangerous phenomenon of conspiracy theory beliefs. We have reworked the discussion to clarify the recommendations and make them more applicable generally for public health.

  • Not just this, but in a subsequent paragraph the authors state: "Science should instead be presenting a story of what its scientific facts mean for an individual, their relatives, their city, and their country. Science, in other words, should heavily invest in marketing and learning from consumer behavior research." This is not an issue with science. This is an issue with science communication or the sociology of science. This is simply bad advice for scientists. Move this discussion on to something more about how we can more effectively communicate the work of scientists to the public, rather than chastise scientists for using appropriate terminology when engaging in scientific study.

Thank you for the advise and we have reworked the section as suggested.

In short: the term "conspiracy theory" needs defining in this paper. The authors also need to do a lot more to describe the survey method, and produce the questionnaire (or, at least, representative sample questions from it). Finally, the authors need to examine some of the claims they make in the article and explain their import more fully.

      We believe we have addressed all these points.

  • Some other notes: Page 2: In the paragraph starting "Additional parameters beyond stress management..." the authors mention factors which might influence or contribute to belief in conspiracy theories. Yet most, if not all, of these factors are also factors leading towards most political beliefs which aren't conspiratorial, so why pick out conspiracy theories as a class of belief here?

We understand the point made by the reviewer and agree that these factors could also lead towards other types of beliefs including political and religious ones. Yet, the factors listed were picked from conspiracy theory bibliography therefore the authors are merely citing the conspiracy theory literature context. Anything beyond that, we believe would confuse readers and take the paper in a different direction. We have reworked the discussion section to make our points clearer.

Reviewer 2 Report

The article raises important and interesting aspects of human behavior during a pandemic COVID-19. Researchers competently presented the results and described them. A significant drawback of the study is due to the fact that the authors did not present a convincing review of the literature on the study topic over the past 5 years. Because of this, their introduction and discussion seem unconvincing and raise a number of questions. In particular, the conclusion that "The findings showcased conspiracy theories are clearly believed even among highly educated individuals" seems to be unfounded data.

The greatest value of the article relates to data on the prevalence of conspiracy theory regarding COVID-19. The conclusions made regarding the relationship of these beliefs with other variables have little scientific novelty.

Author Response

On behalf of the author team, I would like to thank you for giving us the opportunity to revise and resubmit our work. We would also like to thank the reviewers for their comments, which we believe have greatly improved our manuscript.

Below you can find a detailed response to all reviewers’ comments.

Thank you.

Reviewer 2

  • The article raises important and interesting aspects of human behavior during a pandemic COVID-19. Researchers competently presented the results and described them. A significant drawback of the study is due to the fact that the authors did not present a convincing review of the literature on the study topic over the past 5 years.

We have reworked the introduction to present more of a recent review of the literature.

  • Because of this, their introduction and discussion seem unconvincing and raise a number of questions. In particular, the conclusion that “The findings showcased conspiracy theories are clearly believed even among highly educated individuals” seems to be unfounded data.

The reviewer is correct that this conclusion was not clearly explicated. We have reworked both the title and the discussion sections to make this point more clear in relation to the findings of the study.

  • The greatest value of the article relates to data on the prevalence of conspiracy theory regarding COVID-19. The conclusions made regarding the relationship of these beliefs with other variables have little scientific novelty.

We hope that the clarity of the paper and its contributions have improved based on the revisions made.

Round 2

Reviewer 1 Report

This is a much improved paper. My only real suggestion here is some minor revision of the following section:

"Conspiracy theories are commonly defined as beliefs that attempt to explicate uncommon, distressing events, including disease epidemics (Franks, Bangerter, & Bauer, 2013). Usually, conspiracy theories include false accounts attributing cause of unexplained events to human malevolence (Swami, 2012; Dentith, 2018). “Conspiracy theories flourish in times of crisis when people feel threatened, uncertain, and insecure” (Douglas, 2021). Increased anxiety and other mental health problems, such as depression, have been indeed linked to higher levels of conspiracy theory beliefs (Grzesiak-Feldman, 2013). The need to rationalize inexplicable events and diffuse anxiety levels, lead individuals to believing in imagined (Vs. factual) theories (Franks, Bangerter, & Bauer, 2013). Not unlike the ancient Greeks, individuals today form myths and conspiracy theories that entail some truth (e.g., coronaviruses are just another family of viruses with flu like symptoms) and some made-up components (e.g., this coronavirus is just a plot by governments to control us,; Abu Taher, Mannan, Shahinoor, & Dulal, 2018)."

For one, this should be its own paragraph.

For another, it would be good to make explicit the link between myths and conspiracy theories. The authors could use a line like: "In the same way the ancients used myths to explain the world, we might take it that in contemporary society conspiracy theories play the same kind of role."

Yet another thing: the authors write "Usually, conspiracy theories include false accounts attributing cause of unexplained events to human malevolence (Swami, 2012; Dentith, 2018)."

This sentence accurately describes Swami et al.'s views but not Dentith's (Dentith argues that conspriacy theories can be true in a variety of cases, so they are not usually false; Dentith argues this is both a definitional and empirical question, and we do not have enough data to answer it). I would recommend changing this sentence to something like: "Conspiracy theories  attribute the cause of unexplained events to human malevolence (Swami, 2012; Dentith, 2018)."

Author Response

Thank you again for your invaluable contribution that made our paper indeed better and clearer to the readers.

The authors.

Answers to Reviewer

This is a much-improved paper. My only real suggestion here is some minor revision of the following section:

"Conspiracy theories are commonly defined as beliefs that attempt to explicate uncommon, distressing events, including disease epidemics (Franks, Bangerter, & Bauer, 2013). Usually, conspiracy theories include false accounts attributing cause of unexplained events to human malevolence (Swami, 2012; Dentith, 2018). “Conspiracy theories flourish in times of crisis when people feel threatened, uncertain, and insecure” (Douglas, 2021). Increased anxiety and other mental health problems, such as depression, have been indeed linked to higher levels of conspiracy theory beliefs (Grzesiak-Feldman, 2013). The need to rationalize inexplicable events and diffuse anxiety levels, lead individuals to believing in imagined (Vs. factual) theories (Franks, Bangerter, & Bauer, 2013). Not unlike the ancient Greeks, individuals today form myths and conspiracy theories that entail some truth (e.g., coronaviruses are just another family of viruses with flu like symptoms) and some made-up components (e.g., this coronavirus is just a plot by governments to control us,; Abu Taher, Mannan, Shahinoor, & Dulal, 2018)."

For one, this should be its own paragraph.

Thank you. This is now a new paragraph.

For another, it would be good to make explicit the link between myths and conspiracy theories. The authors could use a line like: "In the same way the ancients used myths to explain the world, we might take it that in contemporary society conspiracy theories play the same kind of role."

We really liked this recommendation and added the line as is at the beginning of the paragraph.

Yet another thing: the authors write "Usually, conspiracy theories include false accounts attributing cause of unexplained events to human malevolence (Swami, 2012; Dentith, 2018)."

This sentence accurately describes Swami et al.'s views but not Dentith's (Dentith argues that conspriacy theories can be true in a variety of cases, so they are not usually false; Dentith argues this is both a definitional and empirical question, and we do not have enough data to answer it). I would recommend changing this sentence to something like: "Conspiracy theories  attribute the cause of unexplained events to human malevolence (Swami, 2012; Dentith, 2018)."

Once again, a line that makes our paper better. We followed the recommendation